# Effect of Water Stress and Shading on Lime Yield and Quality

**DOI:** 10.3390/plants12030503

**Published:** 2023-01-21

**Authors:** Ana Belén Mira-García, Wenceslao Conejero, Juan Vera, María Carmen Ruiz-Sánchez

**Affiliations:** Irrigation Department, CEBAS-CSIC, Campus de Espinardo, 30100 Murcia, Spain

**Keywords:** cropping system, lime juice, metabolites, nutrients, shading, soil water deficit

## Abstract

The aim of this study was to test the combined effect of water stress and cropping system on yield and fruit quality in Bearss lime trees. For this purpose, two irrigation treatments were applied during stage II of fruit growth: control (well irrigated, automatically managed by soil water content sensors) and stress (non-irrigated), both under open-field and shaded conditions. Soil water status was assessed by determining soil water content and plant water status by measuring stem water potential (Ψ_stem_), stomatal conductance (g_s_), and net photosynthesis (P_n_). Yield parameters (kg and the number of fruits per tree and fresh mass per fruit) and fruit quality were assessed on two harvest dates. In addition, on the second harvest date, the content of metabolites and nutrients in the lime juice was analyzed. The results showed that soil water deficit induced 35% lower g_s_ values in open-field than in shaded conditions. The highest kg and the number of fruits per tree were observed in the shaded system, especially on the first harvest date. The lowest yield was observed in stressed trees grown without netting. Slightly higher fresh mass and equatorial diameter were observed in shaded fruits than in open-field fruit. Soil water deficit increased fruit total soluble solids and decreased juice content, especially in open-field trees. Shaded conditions made the lime trees more resilient to soil water deficit, which led to higher yields and better external fruit quality traits. In addition, fruit precocity was significantly higher in the shaded system.

## 1. Introduction

Citrus is one of the most extensive woody crops in the world. Citrus fruits are grown in more than 140 countries [1]. In the European Union, Spain is the most important citrus-growing country, with 57% of total production [2]. Citrus production in Spain is concentrated in the Mediterranean area [3], which is characterized by high solar radiation and air temperature, especially during the summer months [4]. Ongoing climate change, which is particularly pronounced in these areas [5], has a negative impact on water availability. This water scarcity context calls for the adoption of cropping techniques and irrigation strategies that increase water use efficiency (WUE).

Shade netting is a cultivation technique that protects crops against biotic and abiotic stresses [6]. The use of netting induces modifications in the environmental conditions surrounding the plants [7] by reducing incoming radiation [8]. This cropping system also promotes an increase in WUE, as was reported in several crops [9,10,11]. In addition, this technique has other benefits related to crop physiological and agronomical aspects. In this regard, Bastías et al. [12] observed an increase of ≈30% in the photosynthetic capacity of blue-shaded apple trees. In addition, improvements in yield and fruit quality have also been observed by Wachsmann et al. [10] in mandarin and by Tinyane et al. [13] in avocados as a consequence of shading conditions. Changes in weather conditions under netting could affect the accumulation of some metabolites and nutrients in plants and fruits. In this sense, Zandalinas et al. [14] observed that an increase in air temperature favors the accumulation of salicylic acid in Carrizo and Cleopatra citrus plants. Jokar et al. [15] reported increases in Ca, P, and N content in Sabz fig fruits grown under blue shade netting.

The application of water stress affects the physiological and production aspects of citrus plants. In this regard, many studies confirm that low water availability negatively affect plant water status, reducing leaf water potential and gas exchange [16,17,18], which is recognized as a tolerance mechanism to cope with drought [19]. Yield and fruit quality traits also depend on soil water availability [20]. Additionally, water deficit affects the content of amino acids, sugars, and organic acids in the fruit, as indicated Morianou et al. [21], who observed an increase of about 10% in the citric acid content of Ruby grapefruit fruits irrigated at 60% of crop water requirements. In Gala apple fruits, Tao et al. [22] observed that the concentration of fructose increased under moderate water deficit conditions. Variations in the nutritional content of fruits could also be the result of reductions in the irrigation dose, as Medyouni et al. [23] observed in deficit-irrigated tomato fruits, which had lower Ca and K content than well-irrigated fruits.

Numerous studies address the effect of shading and deficit irrigation on water relations, yield, and fruit quality of various fruit crops, but they do so separately. To our knowledge, no study has evaluated the combined effect of shading and water stress on physiological and agricultural traits of lime trees, except those of Mira-García et al. [24,25], which focused on soil-plant water status indicators. For these reasons, the main objective of this study was to evaluate the combined effect of cropping system (shading and open-field) and irrigation strategy (control and stress) in Bearss lime trees grown under Mediterranean conditions. Plant water status, yield and fruit quality, including the accumulation of metabolites and nutrients in lime juice, were evaluated to determine the cropping system that would allow lime trees to better cope with water deficits, especially in the current climate change scenario.

## 2. Results

### 2.1. Agro-Meteorological Conditions

The agro-meteorological data recorded during the experimental period (June to September) are shown in Table 1. Monthly maximum air temperatures (T_max_) ranged from 29.2 to 35.7 °C, with slightly higher values in shaded conditions than in open-field. During the experimental period, mean air temperatures (T_mean_) were similar in both growing conditions (≈25.7 °C). Monthly minimum air temperature (T_min_) values were lower in the net house (Table 1). In general, maximum, mean, and minimum relative humidity (RH) values were slightly lower in shaded conditions than in open-field (Table 1). For example, in August, the maximum RH (RH_max_) was 3.3% higher in open-field than in shaded conditions. Monthly vapor pressure deficit (VPD) values ranged from 1.9 to 3.1 kPa, being slightly higher under netting (Table 1). In July and August, rainfall was low; however, at the beginning (June) and at the end of the experiment (September), rainfall was 47.5 and 32.4 mm, respectively. The June rainfall occurred during the first few days, just before the onset of water stress. Crop reference evapotranspiration (ET_0_) monthly values varied throughout the experimental period from 2.5 to 5.5 mm day^−1^ (Table 1). The highest ET_0_ values were observed in open-field conditions, as for solar radiation, with an average difference between the two cropping systems of 21.4 W m^−2^, with the maximum value in July.

### 2.2. Plant Water Status

The values of stem water potential (Ψ_stem_), accumulated water stress integral (SΨ_stem_), stomatal conductance (g_s_) and net photosynthesis (P_n_) obtained at the end of the stress period are shown in Table 2. The Ψ_stem_ values of the control treatments were statistically similar (Table 2), and the same was true for the stressed treatments. Regardless of the cropping system, the stressed treatments showed lower (more negative) Ψ_stem_ values (≈−3 MPa) compared to the control ones (≈−0.9 MPa) (Table 2).

The values of SΨ_stem_ of both control treatments were statistically similar in the trees grown without and with netting (201.5 and 189.5 MPa day, respectively) (Table 2). In contrast, SΨ_stem_ was statistically different in the two stressed treatments (101.9 MPa day higher in open-field). Overall, SΨ_stem_ was significantly higher in the stressed trees than in the control trees, the difference being more pronounced in open-field than in shaded conditions (721.9 and 632.0 MPa day, respectively) (Table 2).

In the control treatments, g_s_ was statistically different, being 35% lower in open-field than in shaded conditions (Table 2). However, no significant differences were observed in g_s_ of the stressed treatments, although it was 14% higher in shaded conditions. In the case of P_n_, non-statistically significant differences were observed between the control treatments and between stress treatments (Table 2). A substantial reduction in both leaf gas exchange parameters was observed in stressed trees with respect to the control values (71 and 76% in open-field and 78 and 65% in shaded, for g_s_ and P_n_, respectively), but without significant differences in either cropping condition (Table 2).

### 2.3. Fruit Yield

Lime fruits were collected on two harvest dates following commercial criteria. A three-way analysis of variance (ANOVA), with harvest date, cropping system, and irrigation treatment as the main factors, revealed no significant effect of harvest date on yield parameters, except for fresh fruit mass (Table 3). The effect of the cropping system on the overall data indicated a significantly higher yield (in terms of kg per tree and fresh fruit mass) under shaded conditions than in open-field. Overall, significantly lower yields were obtained in the stressed trees than in the control trees (Table 3).

The statistically significant interaction between harvest date and cropping system (H × C) was because yield differences between open-field and shaded trees were only significant at harvest I (Figure 1). Furthermore, the interaction between harvest date and irrigation treatment (H × I) was significant because the yield was higher at harvest I than II in the control trees, while the opposite was true for the stressed trees. The harvest date × cropping × irrigation interaction (H × C × I) was not significant (Table 3), so the data for each harvest date were analyzed separately (Figure 1).

At harvest date I, shaded trees produced significantly higher yield (kg and the number of fruits per tree) than open-field trees (Figure 1A,B). For instance, control trees grown under netting yielded 27.4 kg and 354 fruits per tree more than those cultivated in open-field conditions (≈89% increase in both yield parameters). A substantial reduction in yield parameters was observed in stressed trees compared to control trees in both cropping systems at the first harvest date. This fact was especially evident in open-field conditions, where almost no fruit met commercial size requirements (only those for fruit quality measurements were harvested). A different trend was observed in the second harvest date (II), with no significant differences in kg per tree between control and stressed trees grown in open-field or shaded conditions (≈25 kg tree^−1^) (Figure 1A). However, the number of fruits per tree increased significantly in stressed trees compared to control trees (Figure 1B), resulting in lower fresh mass fruit being harvested from stressed trees (Figure 1C).

The total kg of lime per tree (sum of harvest date I and II) was significantly (*p* < 0.01) higher under shaded conditions than in open-field (mean control and stress values of 64.2 and 43.4 kg per tree (Figure 1A), for shaded and open-field, respectively, an increase of 48%), as was mean fresh fruit mass (*p* < 0.05) (74.9, and 63.9 g, in shaded and open-field, respectively, an increase of 17%) (Figure 1C), while there were no significant differences in the total number of fruits per tree (Figure 1B). Water deficits significantly (*p* < 0.001) reduced total kg per tree and mean fruit mass (Figure 1A,C), in stressed trees compared with control trees in both cropping systems (66.4 and 41.1 kg per tree; 84.0 and 54.8 g per fruit in shaded and open-field, respectively), with a similar total number of fruits per tree (Figure 1B). The data for each treatment, as shown in Figure 1A, denote the higher yield values in shaded conditions than in open-field (24% increase in kg per tree and double for control and stress trees, respectively) and highlight the higher total kg per tree and fruit fresh mass obtained in stressed trees grown under shaded conditions compared to those in open-field (Figure 1C).

### 2.4. Lime Fruit Quality

The ANOVA of fruit quality revealed no significant differences between harvest dates in fruit equatorial diameter (Ø_eq_); however, significantly higher values for juice content, skin chroma (*p* < 0.001), total soluble solid content (TSS) (*p* < 0.05), and lower peel thickness (*p* < 0.001) were observed for fruits from harvest II (data not shown).

Lime fruit quality was not strongly affected by the effect of cropping system (C), which was only significant on the first harvest date when higher values of Ø_eq_, juice content, and lower TSS, and maturity index (MI) were observed for shaded fruits than for open-field fruits (Table 4). In contrast, the effect of irrigation (I) was highly significant for all fruit quality traits at both harvest dates, except for titratable acidity (TA) values (Table 4). Regardless of the cropping system, stressed fruits were smaller (lower Ø_eq_) and with lower peel thickness, juice content, and skin chroma at both harvest dates. In contrast, fruit quality parameters such as skin Hue, TSS, and MI increased in the fruits submitted to water stress (Table 4).

When the combined effect of cropping and irrigation was evaluated, slightly higher values (not significant) were observed for most of the studied fruit quality traits in fruits from control trees grown under shaded conditions compared to those grown in open-field. Other internal fruit quality traits, such as TSS and MI, were significantly higher in stressed fruits from trees grown in open-field than in shaded conditions (Table 4). In general, the differences between control and stress treatments in fruit quality traits were even more pronounced in open-field than in shaded conditions (Table 4). Proof of this is that at harvest I, the stressed fruits that were grown in the open-field showed a reduction of 18.9% in Ø_eq_ compared to the control, and this reduction was 4.9% under shaded conditions (Table 4).

### 2.5. Lime Juice Metabolomics Analysis

At harvest II, coinciding with the end of the stress period, the content of amino acids, sugars, and organic acids in the juice of lime fruits from the different treatments was determined. The ANOVA revealed a non-significant effect of the cropping system (C) in most of the metabolites measured, except for the amino acids glutamine and asparagine and lactate and citrate organic acids, which showed the highest values in the open-field grown fruits (Table 5). In contrast, the effect of irrigation (I) was found to be highly significant, increasing the metabolite contents of the stressed fruits compared to the control fruits (Table 5). The interaction between both main factors (C × I) was not significant, except for asparagine (*p* < 0.01).

Metabolites analysis indicated that the main sugars in the lime juice were glucose and fructose, which accounted for an average of 49 and 43% of the total sugars determined, respectively (Table 5). The mean concentration of sucrose in the lime juice was very low, around 20 mM L^−1^, which is about 8% of the total sugars determined (Table 5). As for amino acids, on average, asparagine was the main amino acid in lime juice, with a mean concentration of 33 mM L^−1^, which represents 65% of total amino acid content, followed, in decreasing order, by proline (16%), alanine (10%), glutamine (6%) and 4-aminobutyrate (3%) (Table 5). In lime juice, as expected, the main organic acid was citric acid, with an average concentration of 367 mM L^−1^, while formic and lactic acids were present at concentrations of ≈0.29 Mm L^−1^ (Table 5).

Lime juice from the open-field treatments tended to have higher but non-significant sugar content than juice from the respective shaded treatments (Figure 2A). For example, glucose content was 33% higher in lime juice from control trees grown in open-field than under netting conditions (Figure 2A). The same trend was observed for amino acid and organic acid content (Figure 2B,C).

Sugar content in stressed fruits from both cropping systems was significantly higher than in their control counterparts, with the greatest difference in fruits grown in open-field conditions (Figure 2A). Fruits grown inside the shade net house showed a fructose content in lime juice of 39 and 144 mM L^−1^ for the control and stress treatments, respectively, and 49 and 216 mM L^−1^ for the trees grown in open-field conditions (Figure 2A). The stressed fruits also showed a higher concentration of amino acids and organic acids in their juice compared to the control (Figure 2B,C). In the case of amino acids, the most pronounced differences are in asparagine and proline. These amino acids were significantly higher in the fruits of stressed lime trees grown in open-field than in those grown under shaded conditions (Figure 2B). This was particularly interesting in the case of proline, as the difference between control and stress was 8.7 under shaded conditions and 22.3 mM L^−1^ in open-field conditions. This finding would be related to the level of water stress endured by each treatment, as confirmed by the good exponential relationship found between proline content and the accumulated water stress (Figure 3).

### 2.6. Lime Juice Mineral Content

Nutrient analysis of lime juice from fruits picked at harvest II indicated that K was the main nutrient, with concentrations ranging from 11.9 to 15.1%, depending on the treatment (Table 6). However, the Fe and Mn contents in lime juice were minimal (<1.5‰) (Table 6). The ANOVA reported statistically significant differences as a function of cropping and irrigation effects. There was a tendency for K and Mg to increase and P, Fe, and Mn to decrease in the shaded treatments compared to the open-field treatments, while Ca remained unchanged (Table 6).

The effect of irrigation resulted in lower values for all minerals studied in the well-irrigated fruits (Table 6). This increase in mineral content of the stressed fruits compared to the control fruits was maximal under open-field conditions. For example, the P content was 0.37 and 0.85% higher in stressed fruits than in control fruits, under shaded and open-field conditions, respectively (Table 6). As for K content, higher values were observed in stressed fruits than in control fruits under open-field conditions, while differences between shaded treatments were not significant (Table 6).

## 3. Discussion

### 3.1. Lime Tree Water Status and Fruit Yield

During the experimental period, the weather conditions were typical of the summer season in Mediterranean areas, with a high atmospheric evaporative demand [4]. Inside the shade net house, the microclimate conditions to which the plants were exposed differed slightly from those in the open-field. For example, the maximum air temperature (T_max_) was higher under netting conditions (Table 1). Accordingly, Blakey et al. [26], observed that T_max_ increased by 2.9 °C under white net compared to open-field conditions. Netting conditions also decreased relative humidity (RH) compared to open-field values (Table 1), as reported by Alaphilippe et al. [27], who observed a decrease of 2.3% in mean RH values under Alto-carpo netting. Higher air temperatures and lower RH values led to slightly higher vapor pressure deficit (VPD) values under shaded conditions than in the open field (Table 1).

These changes in the microclimate within the shade modified the physiological and agronomical response of lime trees. In this sense, a slight improvement in the water status of trees grown under shaded conditions was detected, with lower accumulated water stress integral (SΨ_stem_) and higher stomatal conductance (g_s_) values than in open-field conditions (Table 2).

This increase in g_s_ could have promoted higher photo-assimilate production leading to higher lime fruit yield under shaded conditions (Figure 1A). This fact was also suggested by Panigrahi et al. [28], who observed that a 30% increase in g_s_ promoted an improvement in the yield of Kinnow mandarin by 103%. This positive effect of shaded conditions on citrus yield was also reported by Germanà et al. [29] and Wachsmann et al. [10] in Primasole and Ori mandarin trees, respectively, and in other woody crops such as apple [30] and avocado [26]. In our study, the lime yield was higher in shaded trees than in open-field trees, especially at the first harvest date (Figure 1A). This meant that under these shaded conditions, fruit precocity increased, leading to higher market prices.

The water deficit imposed on the lime trees modified plant water status, producing a marked decrease in Ψ_stem_ and leaf gas exchange values in stressed trees compared to control trees (Table 2), similar to that reported in mandarin Conesa et al. [18] and in grapefruit Romero-Trigueros et al. [31]. This decrease in water potential is due to the limitation of water transport from the roots to the leaves as a consequence of low soil water availability conditions. Furthermore, low soil water availability favored leaf stomatal closure to avoid plant dehydration, so that g_s_ was considerably reduced [17,32,33]. The decrease in the stomatal opening has a negative impact on intercellular CO_2_ levels [34], limiting photosynthetic capacity, as corroborated by the decrease in P_n_ observed in stressed lime trees (Table 2).

Deficit irrigation significantly reduced the productive parameters of lime trees (Figure 1), as numerous studies have also shown (e.g., Ballester et al. [35] and García-Tejero et al. [36] in deficit irrigated mandarin and orange trees, respectively). Our results showed that the reduction in total lime fruit yield was a consequence of a decrease in both the number of fruits per tree and fruit size (Figure 1 and Table 4), in agreement with recent studies in orange trees [37]. This could be attributed to the fact that the withholding of irrigation coincided with stage II of active lime fruit growth, the most critical phenological period in terms of water deficit in *Citrus* sp. [38]. During this stage, cell expansion takes place, and water deficit decreases the fruit growth rate, resulting in smaller fruit [39]. The severe water deficit applied to lime trees (Table 2) during the active growth stage resulted in few fruits reaching marketable size, decreasing the number of fruits collected at harvest I (Figure 1B). The similar total number of fruits per tree obtained in the different treatments indicated no significant differences in lime fruit abscission, contrary to the findings of Jamshidi et al. [40], who indicated that yield reduction in deficit irrigated orange trees was attributed to the fruitlet abscission and the lower number of fruits, as deficit irrigation was applied during most of the growing season (from June to January).

Analysis of the combined effect of the cropping system and irrigation treatment revealed that stressed trees grown under shaded conditions behaved differently from those grown in open-field conditions, with higher yield (kg per tree and fresh fruit mass) at harvest I and total (Figure 1A,C). At this point, it should be noted that, although similarly low values for plant water status indicators were observed for stressed trees in the two cropping systems (Ψ_stem_ ≈ −3 MPa, low g_s_ and P_n_) (Table 2), water stress progressed differently, the plant water status indicators values decreased more rapidly in open-field conditions [25]. The lower accumulated water stress integral values measured in shaded trees underline this fact (Table 2). Improvements in leaf gas exchange values under netting as a result of more favorable environmental conditions could have led to an improvement in water use efficiency compared to open-field conditions, as reported in other studies on lemon [9] and apple trees [30].

### 3.2. Lime Fruit Quality

Although the lime fruits met the commercial criteria at both harvest dates (37 days apart), those harvested at the earlier date (harvest I) showed a slightly lower degree of maturity (lower juice content, skin chroma, and total soluble solids) than those harvested at harvest II.

Shaded conditions improved not only kg and number of fruits per tree (Figure 1A,B) but also fruit mass (Figure 1C) and equatorial diameter (Table 4) [41]. Regarding fruit coloration, the cropping system had no significant effect on the color saturation (chroma) of lime fruits. However, a lower degree of yellowness (high Hue angle) was observed for shaded fruits (Table 4). These results were in agreement which those found by Simon-Grao et al. [42], who reported that shade screen produced lemon fruits that were less yellow than those grown in an open field.

No clear differences in juice content were observed between fruits grown in both cropping systems (Table 4), contrary to that observed by Lee et al. [43], who reported a significant increase (6%) in Ponkan mandarin grown under white net conditions compared to fruits cultivated in open-field. Regarding internal fruit maturity, shade conditions decreased the accumulation of sugars in the fruit and delayed the maturation process, as confirmed by lower values of TSS and MI (Table 4). In line with this, Otero et al. [44] reported that shading in late autumn until harvest decreased TSS values in orange juice. However, the cropping system did not affect the acidity of the fruits (Table 4).

When comparing irrigation treatments, significantly lower Ø_eq_, skin chroma, and juice content were observed in stressed fruits than in well-irrigated fruits in both cropping conditions and harvest dates (Table 4). The effect of soil water deficit conditions on fruit quality has been widely reported in deficit-irrigated orange and mandarin trees [45] or peach fruits [46]. In our study, withholding irrigation decreased juice content (Table 4), as reported by Conesa et al. [18], who observed reductions of up to 8% in juice content of deficit irrigated mandarin trees. In parallel, water deficit conditions favored the accumulation of sugars (TSS) in lime juice (Table 4), in agreement with Romero-Trigueros et al. [47] in deficit-irrigated grapefruit trees. Moreover, these reductions advance fruit ripening, as confirmed by the MI data (Table 4).

The combined effect of cropping system–irrigation revealed rather similar fruit chemical characteristics in well-irrigated trees under both shaded and open-field conditions. However, significant differences were found in the case of the stress treatment, when fruits obtained from trees grown under shading were larger (equatorial diameter, juice content) than their counterparts grown in open-field conditions (Table 4). It is also important to note that the differences between stress and control treatments were more pronounced in open-field conditions (Table 4).

### 3.3. Metabolite and Nutrient Contents in Lime Juice

Citrus fruit is a source of many metabolites (sugar, organic acid, and amino acids) that are beneficial to human health [48]. The main sugars are fructose, glucose, and sucrose in a 1:1:2 ratio [49]. The sugar content of fruit juice may depend, among other factors, on the plant species [50] or the stage development of the fruit [51]. In lime fruits, the concentration of sucrose was lower than fructose and glucose (Figure 2A), as also found by Kafkas et al. [52] in Star Ruby grapefruit juice.

In our study, lime juice was also a source of amino acids but in lower concentrations than sugars, as also found by Xie et al. [53] in orange, grapefruit, and lemon juice. The metabolomic analysis revealed that the main amino acids in lime juice were asparagine, proline, alanine, glutamine, and 4-aminobutyrate (Figure 2B), similar to what was reported in orange and lemon juices [54]. The fact that asparagine was the predominant amino acid in lime juice indicates its key role in nitrogen storage and transport in plants [55], as fruits are the main metabolite sink organ.

As expected, the main organic acid in citrus fruits is citric acid [56], which is responsible for the characteristic taste of this species. In the lime juice studied in our experiment, the concentration of citric acid (Figure 2C) was similar to that reported by Penniston et al. [57] in the same species.

Citrus species respond to abiotic and biotic factors by modifying their metabolism. Therefore, the different metabolite contents in lime juice (Figure 2) could be the result of the different environmental conditions in the two cropping systems (Table 1). This idea was confirmed by Zandalinas et al. [14], who observed that increased air temperature favors the accumulation of salicylic acid in Carrizo and Cleopatra citrus plants.

Drought also induces changes in the synthesis of metabolites. In our study, restricted water availability in the soil increased the synthesis of metabolites and favored their accumulation in lime fruits (Figure 2). These results were also observed in Pera sweet orange [58] and in other crops [59,60]. This was especially evident in the case of proline, which increased to a greater extent than the rest of the metabolites in water stress conditions (Figure 2B). This is due to the fact that proline accumulation occurs in response to environmental stress, including water stress [61], and for this reason, it has been defined as a key indicator of plant water deficit [62,63].

It should be noted that for all the metabolites studied, the difference between stress and control treatments was maximal in open-field conditions, which revealed that water deficit was more accentuated in this cropping system.

The response to the combined effect of cropping system-irrigation treatment revealed that the accumulation of metabolites in the lime juice was higher in open-field stressed trees (Figure 2). This could be a consequence of the fact that the rate of metabolite accumulation depends, among other factors, on the irradiation conditions. In open-field, solar radiation was higher than in shaded conditions (Table 1), which could increase the synthesis of metabolites [64]. Moreover, reductions in the soil water content, as experienced in the stress treatment, also further increase the accumulation of metabolites [65].

The main nutrient in lime juice was K (Table 6), as reported in a recent study on key lime [66]. Thus, lime juice is a natural source of K with numerous benefits, which plays a crucial role in maintaining cell function [67].

Shading induced slight changes in the mineral content of lime juice compared to open-field conditions, increasing K and Mg contents and decreasing P, Fe, and Mn contents (Table 6). In this regard, Zhao and Oosterhuis [68] observed that shading increased NO_3_^−^-N and P contents but did not affect the concentration of K in the petiole of cotton. Demirsoy et al. [69] found a lower accumulation of N, P, K, and Ca in the leaves and shoots of strawberry plants grown in open-field conditions. In dried figs, Jokar et al. [15] observed that shading favored the accumulation of Mg, Ca, and P but had no effect on K.

Soil water availability has a positive effect on the nutritional composition of fruits. In this study, water stress increased the accumulation of several nutrients (Ca, P, K, Fe, Mg, and Mn) in lime juice (Table 6). This could be the result of a smaller fruit size in stressed trees compared to control trees, in agreement with what has been observed in other citrus species by Ballester et al. [70]. This decrease in fruit size led to lower water content [71] and a nutrient concentration effect.

## 4. Materials and Methods

### 4.1. Experimental Conditions

The experiment was conducted in the 2021 fruit growing season, in a 0.8 ha plot, at the experimental field station of CEBAS-CSIC in Murcia, Spain (38°06′31” N, 1°02′14” W, 110 m above sea level). The plant material consisted of five-year-old Bearss lime trees (*Citrus latifolia* Tan.), grafted onto *Citrus macrophylla* L. rootstocks and planted on ridges at 6 m × 4 m tree spacing. The ridges were shaped using soil already present in the orchard with a rear V-blade attached to a tractor. The ridges were 2 m (bottom) and 1.5 m (top) wide and 0.4 m high. The trees were irrigated with a double drip line, 1 m apart, and located on both sides of the tree trunks. Each drip line contained two pressure-compensated emitters of 4 L h^−1^, located 0.7 m from the tree trunk.

The soil was stony and highly calcareous, with a clay loam texture (40% sand, 28% loam, and 32% clay), average bulk density of 1.43 g cm^−3^, and organic matter content of 1.4%. The volumetric soil water content (θ_v_) at field capacity (FC), and at permanent wilting point (WP) were 0.33 and 0.14 m^3^ m^−3^, respectively [72]. Fertilization, weed, and pest control were carried out following the local practices of citrus farmers.

#### 4.1.1. Cropping Systems

The lime trees were grown in two cropping systems from planting:(i)Open-field: trees grown in direct sunlight, as the most common cropping system used for citrus crops in Mediterranean environments.(ii)Shaded: trees grown inside a shade screen house (60 m length × 40 m wide and 4.5 m high). The HDP white agricultural net used had a light transmission of 76%. A detailed description is provided by Mira et al. [25].

#### 4.1.2. Irrigation Treatments

During the experimental period (day of the year (DOY) 158 to 267), two irrigation treatments were established in each cropping system:(i)Control treatment: trees were irrigated to meet their water requirements throughout the experimental period. Irrigation was scheduled based on real-time volumetric soil water content (θ_v_) threshold values, measured with TDR-type sensors (model 315H, Acclima Inc., Meridian, ID, USA). The sensors were installed horizontally 0.1 m from the emitter near the tree trunk at 0.2, 0.4, 0.6, and 0.8 m soil depths by soil excavation in two representative lime trees of each cropping condition. Irrigation started when mean values of θ_v_ in the active root zone (0.2 to 0.6 m soil depths) reached 30% of the available soil water content (calculated as the difference between FC and WP) and ended when θ_v_ reached the FC value in these average soil layers. Irrigation was activated and stopped remotely by opening and closing solenoid valves using a telemetry system, which read the θ_v_ values every 5 min and recorded 15 min averages (ADCON Telemetry, Klosterneuburg, Austria). Trees in both cropping systems were irrigated equally, as θ_v_ values were considered as the mean of four monitored trees (two in shaded conditions and two in open-field conditions).(ii)Stress treatment: irrigation was suspended during the phenological period corresponding to stage II of lime fruit growth, which in 2021 lasted from 7 June to 24 September 2021 (DOY 158-267). Before the experimental period, all trees were irrigated in the same way as the control.

### 4.2. Measurements

#### 4.2.1. Agro-meteorological Conditions

During the experimental period, agro-meteorological variables, including air temperature (T), relative humidity (RH), solar radiation, wind-speed, and rainfall, were monitored continuously in open-field and shaded conditions with two automatic weather stations. The 15 min average data were transmitted to the aforementioned cloud-based web server platform. The daily average vapor pressure deficit (VPD) was calculated for each cropping condition from the daily maximum (T_max_) and minimum RH (RH_min_) data, as indicated in Allen et al. [73]. Crop reference evapotranspiration (ET_0_) values were calculated hourly using the Penman–Monteith equation [73].

#### 4.2.2. Plant Water Status

During the experiment, water status of lime trees was determined by measuring leaf gas exchange (net photosynthesis (P_n_) and stomatal conductance (g_s_)) and stem water potential (Ψ_stem_) in fully expanded leaves (in four trees per treatment, one from each replication, n = 4) every 10–15 days. Leaf gas exchange was measured around 08:00 solar time in sun-exposed leaves with a portable photosynthesis system (model LI-6400, LI-COR, Lincoln, NE, USA) equipped with a broad-leaf chamber (6 cm^2^) and CO_2_ injector. Measurements were performed at a constant light intensity (≈1200 µmol m^−2^ s^−1^) provided by the red and blue LED source mounted in the leaf chamber. The CO_2_ concentration and airflow rate inside the chamber were maintained at ≈400 µmol mol^−1^ and 350 µmol s^−1^, respectively. Ψ_stem_ was measured on the same trees used for the leaf gas exchange measurements at midday (around 12:00 solar time) using a pressure chamber (model 3000, Soil Moisture Equipment Corp., Goleta, CA, USA) and following the recommendations of Hsiao [74], in leaves from the north side of the trees and close to the tree trunk that were bagged with foil-covered aluminum envelopes at least 2 h before the measurements to limit transpiration [75]. The water stress integral (SΨ_stem_) was calculated from the equation defined by Myers [76]:(1)$${S\Psi }_{\mathrm{stem}}=\sum^{\;}({\bar{\Psi }}_{i,i+1}-\Psi_{c})\cdot n$$
where, ∑ is the sum of the Ψ_stem_ measurements; $\bar{}$
_i,i + 1_ is the mean Ψ_stem_ for any measurement, _i_ and _i + 1_; Ψ_c_ is the maximum Ψ_stem_ value measured during the experiment; and n is the number of days of the experimental period.

#### 4.2.3. Yield and Fruit Quality

Lime fruits were collected on two harvest dates (I): 18 August 2021 (DOY 230) and (II): 24 September 2021 (DOY 267, coinciding with the end of stress period). On these two dates, the weight (kg) and number of limes per tree were determined on four trees per treatment, one from each replication. Total yield was weighed with an electronic scale (Model SSH 92, 0–6000 ± 2 g, Scaltec, Dania Beach, FL, USA), and the number of fruits per tree was counted manually in the field. Fresh fruit mass (g) was calculated as the ratio between kg and number of fruits per tree.

A subsample of 25 fruits was randomly selected from each experimental tree and taken to the lab in isolated boxes to determine fruit quality parameters. Fruit weight was determined using a portable scale (model PE360, Mettler, Aldaia, Spain), and equatorial diameter (Ø_eq_) and peel thickness were measured with a digital caliper (model CD-15D, Mitutoyo, Sakado, Japan). The external fruit color was measured with a Konica Minolta Chroma Meter (model CR-10, Osaka, Japan), and the results were expressed in CIELAB color space, with the chromatic coordinates: L* (lightness), a* (red–green component), b* (blue–yellow component). From these values, the chroma or chromaticity (C*) and Hue angle or tone (h°) were calculated as
(2)$$C^{*}=\sqrt{{\left(a^{*}\right)}^{2}+{\left(b^{*}\right)}^{2}}$$
h° = arc tan (b*/a*)
(3)



The fruits were then squeezed using a common juice extractor, and peel was weighed to determine the juice content (Juice %), as the difference between fruit and peel weights. Total soluble solids (TSS) and titratable acidity (TA) were determined using a Brix/Acidity meter (Atago PAL-BX/ACID F5 Master Kit- Multifruits, Tokyo, Japan). Maturity index (MI) was calculated as the ratio between TSS/TA.

#### 4.2.4. Metabolomics and Nutrients Analysis

On the second harvest date, on 24th September 2021, metabolite and nutrient contents in the lime juice of the fruits sampled for quality analysis were assessed. Juice samples were centrifuged at 10,000 rpm for 30 min at a temperature of 4 °C, and the supernatant was filtered through a 0.45 µm chromatography filter. The content of primary metabolites (sugar, amino acids, and organic acids) and nutrients in these juice samples were then analyzed. The concentration of primary metabolites was quantified by H NMR spectra, recorded at 298 K on a Bruker AVIII HD 500 NMR spectrometer (500.13 MHz for 1H) equipped with a 5 mm CPP BBO cryogenic probe (Bruker Biospin, Ettlingen, Germany). Analysis of metabolites was assessed by the Metabolomics Platform of CEBAS-CSIC (http://www.cebas.csic.es/general_english/metabolomics.html (accessed on 24 September 2021)), while mineral content was assessed by the Ionomics Service of CEBAS-CSIC (http://www.cebas.csic.es/general_english/ionomics.html (accessed on 24 September 2021)), using Inductively Coupled Plasma (ICP-ICAP 6500 DUO Thermo, Horsham, England).

### 4.3. Experimental Design and Statistical Analysis

For each cropping system (consisting of a sub-plot of 60 lime trees each), the control and stress treatments were distributed in a completely randomized design, with four replications, each consisting of one row of six trees for each cropping system, with a total of 24 trees. One tree from each replication was used for measurements, and the rest were considered as borders. Data were subjected to three-way analysis of variance (ANOVA) using SPSS v.25 (IBM, Armonk, NY, USA). Statistical comparisons were considered significant at a probability level (*p*) < 0.05. Post hoc pair-wise comparison between all means was performed by the Least Significant Difference (LSD) multiple range test at *p* < 0.05. Relationships between accumulated water stress integral (SΨ_stem_) and proline content were fitted to an exponential regression.

## 5. Conclusions

The combined effect of cropping system (open-field and shading) and irrigation treatment (well irrigated and non-irrigated during fruit growth stage II) in a lime orchard grown under Mediterranean conditions revealed that net shading conditions mitigate the effects of water stress on yield and fruit quality. Despite only one growing season of study, the results obtained seem sufficiently sound and robust to support the conclusions.

The higher accumulated water stress in the open field had a detrimental effect on yield traits, with significantly lower kg and the number of fruits per tree than under shading, especially in the first harvest, with this earliness meaning better fruit prices and a more profitable crop under shading.

Lime fruit quality was slightly affected by the cropping system; shading increased the fruit size and delayed coloration change, while sugar accumulation and maturity were promoted in the open-field. Regardless of the cropping system, withholding irrigation led to a significant decrease in fruit size and juice content and an increase in total soluble solids and fruit maturation in stressed trees compared to control trees.

The cropping system and irrigation treatment affected the metabolite and nutrient content of lime juice differently. Thus, the accumulation of sugars, amino acids, and organic acids increased in the open-field, while nutrient enrichment was observed in the shade net house. Soil water stress led to an increase in metabolites and nutrient content of lime fruits in both cropping conditions.

Crop shading is proposed as a useful tool in the face of ongoing climate change, especially in areas with scarce water resources, such as semi-arid irrigated zones.

## Data Availability

Not applicable.

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
