# Peer review of "Effect of Water Stress and Shading on Lime Yield and Quality"

_plants, 2023, doi:10.3390/plants12030503_

Round 1

Author Response

Accepted with minor Revisions

This paper deals with the combined effect of cropping system (shading and open field) and irrigation treatment (control and stress) in lime trees, with agronomical (yield and fruit quality) and physiological (water relations) approaches. The subject clearly fits into the special issue aims “Crop-Water Relations: Improving Water Use Efficiency in a Changing Climate”, since the main objective of this paper is to determine which cropping system enhance water use efficiency and enable trees to better cope with water deficits, in the current climate change scenario.

Also, to be noted that no too many work address the differentiate response of shading and water deficits together with the combined effects of both factors.

The weakness of the work is that only one growing season data is showing. Even so, the results obtained in this study are sound enough to support the conclusions.

However, some aspects need to be clarified before publication:

           Firstly, we would like to thank you for the effort and compromise you have showed revising this work. All the comments and suggestions have clearly improved the manuscript, and we hope these changes make the paper suitable for publication in your journal.

- Line 1: I suggest changing the title to: Effects of water stress and shading on lime yield and quality.

            Following your advice, the title has been changed.

Abstract

- Line 14-15: I suggest a comprehensive rephrasing of these lines, as metabolomic and nutrient content in lime juice only was analyzed in the second harvest and not in both harvest dates.

            Done the sentence now reads: “Additionally, on the second harvest date, the content of metabolites and nutrients in the lime juice was analyzed.”

Results

- Line 96: Do the authors have agro-meteorological parameters apart from temperature and relative humidity? I suggest including solar radiation and ETo data in Table 1, if available. This will help readers to better understand the differences in climatic conditions between shading and open-field.

            Data of solar radiation and reference crop evapotranspiration (ET0) have been included in Table 1. A new sentence including comments of these results has been added in the Results section: “Crop reference evapotranspiration (ET0) varied throughout the experimental period from 2.5 to 5.5 mm day-1. The highest ET0 values were observed in open-air conditions, with a maximal difference between the two cropping systems of 1.8 mm day-1. Solar radiation was higher in open-field than in shaded conditions, with a mean difference between both cropping systems of about 21.4 W m-2, been maximal in July.”  

- One decimal place is enough in Table 1 (meteo data).

            Done, only one decimal place is shown.

-Line 101: The table 2 also refers to the accumulated water stress integral data. I suggest including this information in these lines.

            “Accumulated water stress integral” has been included in the sentence.

- Line 110-111: These lines refer to leaf gas exchange parameters in the plural, but the authors only refer to stomatal conductance. What about net photosynthesis?

            Net photosynthesis was also reduced. The paragraph referring to leaf gas exchange has been rewritten and the sentence now reads: “In the case of Pn, non-statistically significant differences were observed between the control treatments, and also between stress treatments (Table 2).”

- Lines 162: Instead of using absolute values for differences between treatments, the authors should indicate percentages (increase or decrease), what 66 kg supposed?

            The figures meaning percentages of increase have been included in text for yield data. The sentences now read: “Total lime fruit yield (the sum of harvest date I and II) was significantly (p < 0.01) higher in shaded than in open-field conditions (average values 64.2 and 43.4 kg per tree for shaded and open-field, respectively, 48% increase), as well as mean average fruit weight mass values (p < 0.05) (74.9, and 63.9 g, in shaded and open-field, respectively, 17% increase), ……….Data for each treatment, as shown in Figure 1, denote the higher yield values in shaded than in open-field conditions (24% increase in kg per tree and double for control and stress trees, respectively)”

- Line 243: the same for proline content. Indicate percentage

            The figures for the % of increase are too big. Data on figure 2 are clear enough to represent that water stress induced increase in the proline content with respect to control treatment, and that this increase was greater in open-field than in shaded conditions.

- Lines 169, 249 and 255: I suggest that the symbols used for the different treatments be included in the caption of each figure.

            The symbols along with the abbreviations denoting each treatment have been included in the caption to Figures 2 and 3.

- Line 276: Please, change the units of Table 6 to SI units (% is not SI)

            The % is dimensionless.

Discussion

- The results in Table 4 show slightly high skin hue values in fruits grown in shaded conditions. I think this indicates that fruits showed a lower yellowness than those grown in open-field. Nevertheless, the authors claim the opposite. Please revise the lines 351-352 in the discussion section and the lines 570-571 in the conclusions section.

            You are right, higher skin hue value implies higher greenness. Now the sentence reads: “Nevertheless, a lower degree of yellowness (high Hue angle) was observed for the shaded fruits (Table 4)”

- Lines 381-431: A more comprehensive discussion of the combined effect of cropping system and irrigation treatment in the accumulation of metabolites and nutrients should be included.

            Done. This part of the discussion has been rephrased to improve reading.

Materials and methods

- Line 506: Which Ystem value is used in the accumulated water stress equation? Was it the same for both cropping conditions? Please indicate.

            In the accumulated water stress equation, Ψc i.e.  the maximum Ψstem value measured during the experiment was -0.52 MPa, in the both cropping conditions.

- Line 525: Chroma instead of chrome.

            Chroma has been corrected throughout the manuscript

- Line 527: The square root symbol for Chroma equation does not look clear.

            The equation has been rewritten using the MS-Word notation

Reviewer 2 Report

This study provides detailed discussions about the combined effect of irrigation and cropping systems on yield and fruit quality in lime trees. The manuscript is well written with proper research design, analysis, and discussion, and would be of interest to the readers as it provides useful information about the lime trees. Nevertheless, I have a couple of concerns that I am sharing here.

my major concern is about the experiment period which only took place for one season. While the trees are young (5 years old) and may reflect the impact of imposed stress in one year, it remains unknown if the next season(s) response of the trees would be the same. would the trees recover from the water stress or do they carry over the impact of the imposed stress? Addressing these types of questions requires more experiments (more than one season). Perhaps, the authors could state this limitation in the abstract or conclusion that the results are only based on one year of study (or if they have other data/information, they could include it in the paper).

I suggest using more quantitative language in the abstract. For example, instead of “no irrigation”, the authors could state the stress percentage (with respect to the total water requirement). Or, instead of:

“The results showed that soil water deficit induced lower gs values in open-field than in shaded conditions” à quantify how much was the difference between the two treatments?

What was the level of applied water stress compared to the fully irrigated trees? According to the M&M, the authors applied regulated deficit irrigation (the water stress applied during the phenological period).

Did rainfall impact the treatments (irrigated versus non-irrigated in the open field and perhaps shaded condition)?

I wonder if the stressed trees could compensate for the reduced stomatal conductance at any time during the applied water stress period. In other words, did the physiological responses of the stressed trees were accumulated during the non-irrigation period? Or was there any response to other environmental variables such as VPD?

I suggest the authors provide detailed information about the observed physiological parameters including net photosynthesis, stomatal conductance, and stem water potential as a time-series plot or something similar (in addition to table 2 which shows the parameters at the end of the stress period). The measurements were done at 10-15 days intervals (line 493) and providing such information about the differences between the treatments would add more value to the manuscript.

Line 317-329. The authors attributed the fruit size and number to the water stress level. I wonder if the reduced number of fruits was the result of the fruitlet abscission, or if the fruit did not reach maturity for harvest? For example, in this study on orange trees (
https://doi.org/10.1007/s00271-020-00709-9), the authors applied DI irrigation during most of the growing season and most of the yield reduction was attributed to the fruitlet abscission. Please compare your results with the mentioned and discuss how regulated deficit irrigation may change fruitlet abscission compared to longer water stress application.

In addition to the impact of irrigation, I wonder if the authors considered the role of VPD on the trees' physiological response and yield by comparing the open-field versus netting cropping systems. In a number of studies on citrus trees (e.g., https://doi.org/10.1007/s00271-017-0562-8,  https://doi.org/10.1016/j.agwat.2019.105838,) the role of VPD has been investigated as an important factor on regulating stomatal conductance (in conjunction with soil moisture). It would be beneficial to discuss this aspect of the work in the discussion section.

I suggest explaining the abbreviations in the figures’ captions. For example, Figure 1: OF stands for Field and etc.

Author Response

This study provides detailed discussions about the combined effect of irrigation and cropping systems on yield and fruit quality in lime trees. The manuscript is well written with proper research design, analysis, and discussion, and would be of interest to the readers as it provides useful information about the lime trees. Nevertheless, I have a couple of concerns that I am sharing here.

            Firstly, we would like to thank you for the effort and compromise you have showed revising this work. All the comments and suggestions have clearly improved the manuscript, and we hope these changes make the paper suitable for publication.

- My major concern is about the experiment period which only took place for one season. While the trees are young (5 years old) and may reflect the impact of imposed stress in one year, it remains unknown if the next season(s) response of the trees would be the same. would the trees recover from the water stress or do they carry over the impact of the imposed stress? Addressing these types of questions requires more experiments (more than one season). Perhaps, the authors could state this limitation in the abstract or conclusion that the results are only based on one year of study (or if they have other data/information, they could include it in the paper).

             The purpose of this study was to evaluate the effects of water deficits applied during the fruit growth period (stage II) on lime yield and fruit quality, and the combined response to shading conditions. Trees were well irrigated before and after that period. Thus, only the current year’ s harvest will be affected by the water stress, and no carry over effect is expected as it is the case in fruit trees, where floral differentiation processes coincide with the current year’s harvest. Also, even though it comprises a single-year data, there are evidences that the results seem sound and robust enough for supporting the conclusions. Following your advice a sentence has been included in the Conclusion: “In spite of just one growing season of study, the obtained results seem sound and robust enough for supporting the conclusions”.

- I suggest using more quantitative language in the abstract. For example, instead of “no irrigation”, the authors could state the stress percentage (with respect to the total water requirement).

       In the experimental conditions “no irrigation” meant irrigation withholding during the fruit growth period. Therefore, this supposed 0% of total water requirements.

- Or, instead of: “The results showed that soil water deficit induced lower gs values in open-field than in shaded conditions” à quantify how much was the difference between the two treatments?

            The difference is indicated in Line 110: “The results showed that soil water deficit induced 35% lower gs values in open-field than in shaded conditions”

- What was the level of applied water stress compared to the fully irrigated trees? According to the M&M, the authors applied regulated deficit irrigation (the water stress applied during the phenological period).

            As above indicated, the experiment was not a conceptual RDI strategy, since irrigation was suspended at all during the period corresponded to the active fruit growth stage. We only mentioned in the discussion section that this is a critical period for applying RDI strategies. In order to avoid misunderstanding all comments to regulated deficit irrigation RDI strategies has been removed from the manuscript.

- Did rainfall impact the treatments (irrigated versus non-irrigated in the open field and perhaps shaded condition)?

            As indicated in Lines 90-91 rainfall events were rare and it appears at the beginning of the June (before the start of irrigation withholding). No rainfall amount differences were observed between open-field and shaded conditions.  

- I wonder if the stressed trees could compensate for the reduced stomatal conductance at any time during the applied water stress period. In other words, did the physiological responses of the stressed trees were accumulated during the non-irrigation period? Or was there any response to other environmental variables such as VPD?

            The physiological response of water deficits was deeply studied in a previous published paper by Mira et al. (2022): (Water status and thermal response of lime trees to irrigation and shade screen. Agricultural Water Management 272, 107843. https://doi.org/10.1016/j.agwat.2022.107843). In this study you will find information on the course of water stress accumulated in the different conditions, as well as the response to environmental variables.

- I suggest the authors provide detailed information about the observed physiological parameters including net photosynthesis, stomatal conductance, and stem water potential as a time-series plot or something similar (in addition to table 2 which shows the parameters at the end of the stress period). The measurements were done at 10-15 days intervals (line 493) and providing such information about the differences between the treatments would add more value to the manuscript.

            In the Lines we indicated “… that, even though similarly low values were noted for plant water status indicators in stressed trees in the two cropping systems (Ψstem ≈ -3 MPa, low gs and Pn), the water stress progressed differently, the plant water status indicators values decreasing more quickly in open-field conditions [27].” As above indicated, more information on behavior of the physiological variables studied can be found in the previous paper by Mira et al. (2022).

- Line 317-329. The authors attributed the fruit size and number to the water stress level. I wonder if the reduced number of fruits was the result of the fruitlet abscission, or if the fruit did not reach maturity for harvest? For example, in this study on orange trees (https://doi.org/10.1007/s00271-020-00709-9), the authors applied DI irrigation during most of the growing season and most of the yield reduction was attributed to the fruitlet abscission. Please compare your results with the mentioned and discuss how regulated deficit irrigation may change fruitlet abscission compared to longer water stress application.

            Our experiment is not a deficit irrigation strategy but a withholding irrigation treatment. As above indicated, the period of no-irrigation (water deficit) start in early June, which in Mediterranean climate in the North-hemisphere coincides with the stage II of fruit growth for most Citrus species, and comprises the elongation cellular processes. The main fruitlet abscission process take place at the end of stage I. Nevertheless, the fruitless abscission observed was very low and affected similarly to both control and stress treatments. Also, from data in Figure 1 it was clear that a similar total number of fruits per tree was recorded in the different cropping systems and irrigation treatments, then, most of the yield reduction was attributed to the fruitlet abscission.

 - In addition to the impact of irrigation, I wonder if the authors considered the role of VPD on the trees' physiological response and yield by comparing the open-field versus netting cropping systems. In a number of studies on citrus trees (e.g., https://doi.org/10.1007/s00271-017-0562-8 y /10.1016/j.agwat.2019.105838) the role of VPD has been investigated as an important factor on regulating stomatal conductance (in conjunction with soil moisture). It would be beneficial to discuss this aspect of the work in the discussion section.

            After reading the interesting papers you recommended, we inform you that the impact of VPD on the physiological response to water deficit has been studied in the previous published paper (above indicated) by Mira et al (2022). Other effect of shading on leaf gas exchange has been also deeply studied in the study of Mira et al. (2020) (Leaf water relations in lime trees grown under shade netting and open-air. Plants (MDPI) 9(4): 510; https://doi.org/10.3390/plants9040510), who indicated that the stomata remained open longer during the day in shaded conditions.

            We consider that in the present manuscript the values of stem water potential and leaf gas exchange inform us on the degree of water stress trees supported, but the study focusses on the combined effect of cropping system (shading vs. open-field) and irrigation (well irrigated vs. water stress) on lime fruit yield and physical and chemical fruit quality (included the analysis of metabolites and nutrients on the lime juice). There are profuse references addressing the effect of shading and deficit irrigation on water relations, yield and fruit quality of various fruit crops, but do so separately. Our study addresses the combined effect of both factors.

- I suggest explaining the abbreviations in the figures’ captions. For example, Figure 1: OF stands for Field and etc.

            Done. The legend, with symbols and abbreviations, has been included in the caption to figures 1, 2 and 3.

Reviewer 3 Report

This manuscript describes the impact of a combined effect of a limited irrigation and shaded lime trees on yield and quality parameters.

The document is well written, some minor corrections, mainly form are detailed in the document attached

Author Response

We would like to thank you for the effort and compromise you have showed revising this work. All the comments and suggestions have clearly improved the manuscript, and we hope these changes make the paper suitable for publication.

- Table 3 please indicate what statistical letters mean in the footnote also

            Corrected.

- Figure 1 Fruit mass. here is not the total is the mean of the two harvest?

            Corrected. TOTAL applied for kg and number of fruit per tree, while MEAN for Fresh fruit mass.

- L-208: amino acids

            Done

- L-210: fruits

            Done

- Figure 2 and 3: please indicate in this figure and the previous one in the legend what OF and S mean

            Done. The legend, with symbols and abbreviations, has been included in the caption to figures 1, 2 and 3.

- L- 567: yield?

            Done. The sentence now reads: “Lower yield (kg and number of fruits per tree)…”

Round 2

Reviewer 2 Report

I don’t have any concerns about this manuscript. the authors have adequately responded to my comments. I would only suggest that the authors summarize some of their responses with reference to the question as points of discussion in their work (if already not included).

I wonder if the stressed trees could compensate for the reduced stomatal conductance at any time during the applied water stress period. In other words, did the physiological responses of the stressed trees were accumulated during the non-irrigation period? Or was there any response to other environmental variables such as VPD?

            The physiological response of water deficits was deeply studied in a previous published paper by Mira et al. (2022): (Water status and thermal response of lime trees to irrigation and shade screen. Agricultural Water Management 272, 107843. https://doi.org/10.1016/j.agwat.2022.107843). In this study you will find information on the course of water stress accumulated in the different conditions, as well as the response to environmental variables.

- Line 317-329. The authors attributed the fruit size and number to the water stress level. I wonder if the reduced number of fruits was the result of the fruitlet abscission, or if the fruit did not reach maturity for harvest? For example, in this study on orange trees (https://doi.org/10.1007/s00271-020-00709-9), the authors applied DI irrigation during most of the growing season and most of the yield reduction was attributed to the fruitlet abscission. Please compare your results with the mentioned and discuss how regulated deficit irrigation may change fruitlet abscission compared to longer water stress application.

            Our experiment is not a deficit irrigation strategy but a withholding irrigation treatment. As above indicated, the period of no-irrigation (water deficit) start in early June, which in Mediterranean climate in the North-hemisphere coincides with the stage II of fruit growth for most Citrus species, and comprises the elongation cellular processes. The main fruitlet abscission process take place at the end of stage I. Nevertheless, the fruitless abscission observed was very low and affected similarly to both control and stress treatments. Also, from data in Figure 1 it was clear that a similar total number of fruits per tree was recorded in the different cropping systems and irrigation treatments, then, most of the yield reduction was attributed to the fruitlet abscission.

 - In addition to the impact of irrigation, I wonder if the authors considered the role of VPD on the trees' physiological response and yield by comparing the open-field versus netting cropping systems. In a number of studies on citrus trees (e.g., https://doi.org/10.1007/s00271-017-0562-8 y /10.1016/j.agwat.2019.105838) the role of VPD has been investigated as an important factor on regulating stomatal conductance (in conjunction with soil moisture). It would be beneficial to discuss this aspect of the work in the discussion section.

            After reading the interesting papers you recommended, we inform you that the impact of VPD on the physiological response to water deficit has been studied in the previous published paper (above indicated) by Mira et al (2022). Other effect of shading on leaf gas exchange has been also deeply studied in the study of Mira et al. (2020) (Leaf water relations in lime trees grown under shade netting and open-air. Plants (MDPI) 9(4): 510; https://doi.org/10.3390/plants9040510), who indicated that the stomata remained open longer during the day in shaded conditions.

            We consider that in the present manuscript the values of stem water potential and leaf gas exchange inform us on the degree of water stress trees supported, but the study focusses on the combined effect of cropping system (shading vs. open-field) and irrigation (well irrigated vs. water stress) on lime fruit yield and physical and chemical fruit quality (included the analysis of metabolites and nutrients on the lime juice). There are profuse references addressing the effect of shading and deficit irrigation on water relations, yield and fruit quality of various fruit crops, but do so separately. Our study addresses the combined effect of both factors.

Author Response

"I don’t have any concerns about this manuscript. the authors have adequately responded to my comments. I would only suggest that the authors summarize some of their responses with reference to the question as points of discussion in their work (if already not included).

            Again, we would like to thank you for the effort and compromise you have showed revising this work. A new paragraph has been included and English language has been checked deeply.

  - I wonder if the stressed trees could compensate for the reduced stomatal conductance at any time during the applied water stress period. In other words, did the physiological responses of the stressed trees were accumulated during the non-irrigation period? Or was there any response to other environmental variables such as VPD?

            The physiological response of lime trees water déficits, including to VPD changes was deeply studied in a previous published paper by Mira et al. (2022), In this study you will find information on the course of water stress accumulated in the different conditions, as well as the response to environmental variables. The paper is included in the list of references

- Line 317-329. The authors attributed the fruit size and number to the water stress level. I wonder if the reduced number of fruits was the result of the fruitlet abscission, or if the fruit did not reach maturity for harvest? For example, in this study on orange trees (https://doi.org/10.1007/s00271-020-00709-9), the authors applied DI irrigation during most of the growing season and most of the yield reduction was attributed to the fruitlet abscission. Please compare your results with the mentioned and discuss how regulated deficit irrigation may change fruitlet abscission compared to longer water stress application.

           A paragraph have been included in LInes 342-346 of the Discusion: “The similar total number of fruits per tree obtained in the different treatments indicated no significant differences in lime fruit abscission, contrary to the findings of Jamshidi et al. [40], who indicated that yield reduction in deficit irrigated orange trees was attributed to the fruitlet abscission and the lower number of fruits, as deficit irrigation was applied during most of the growing season (from June to January).”